# Novel Crosstalks between Circadian Clock and Jasmonic Acid Pathway Finely Coordinate the Tradeoff among Plant Growth, Senescence and Defense

**DOI:** 10.3390/ijms20215254

**Published:** 2019-10-23

**Authors:** Yuanyuan Zhang, Cunpei Bo, Lei Wang

**Affiliations:** 1Key Laboratory of Plant Molecular Physiology, CAS Center for Excellence in Molecular Plant Sciences, Institute of Botany, Chinese Academy of Sciences, Beijing 100093, China; zhangyy@ibcas.ac.cn (Y.Z.); bcp1984@163.com (C.B.); 2University of Chinese Academy of Sciences, Beijing 100049, China

**Keywords:** circadian clock, jasmonic acid, crosstalk

## Abstract

Circadian clock not only functions as a cellular time-keeping mechanism, but also acts as a master regulator to coordinate the tradeoff between plant growth and defense in higher plants by timing a few kinds of phytohormone biosynthesis and signaling, including jasmonic acid (JA). Notably, circadian clock and JA pathway have recently been shown to intertwine with each other to ensure and optimize the plant fitness in an ever-changing environment. It has clearly demonstrated that there are multiple crosstalk pathways between circadian clock and JA at both transcriptional and post-transcriptional levels. In this scenario, circadian clock temporally modulates JA-mediated plant development events, herbivory resistance and susceptibility to pathogen. By contrast, the JA signaling regulates clock activity in a feedback manner. In this review, we summarized the cross networks between circadian clock and JA pathway at both transcriptional and post-transcriptional levels. We proposed that the novel crosstalks between circadian clock and JA pathway not only benefit for the understanding the JA-associated circadian outputs including leaf senescence, biotic, and abiotic defenses, but also put timing as a new key factor to investigate JA pathway in the future.

## 1. Introduction

Circadian clock, an internal timekeeping mechanism, regulates plant growth and development by synchronizing the internal biological and physiological events with the external daily light-dark cycle, thus to enhance fitness of plants [1,2]. The circadian clock molecular system is majorly composed of three parts, namely the input pathways, core oscillator, and output pathways. The input pathways can perceive and recognize the rhythmic environmental cues, then transfer the external timing information to core oscillator through entraining mechanism. Conceptually, the self-sustained central oscillator is based on a series of transcriptional-translational feedback loops. The central loop of core oscillator is formed by the reciprocally repression between TIMING OF CAB EXPRESSION 1 (TOC1) which is the founding member of the PSEUDO-RESPONSE REGULATOR (PRR) family, and CIRCADIAN CLOCK-ASSOCIATED 1 (CCA1)/LATE ELONGATED HYPOCOTYL (LHY), two MYB-domain containing transcriptional repressors, with the expression peak at dawn [3,4]. In the morning loop, CCA1/LHY can repress the expression of *PRR7* and *PRR9*, by directly binding to the evening element (EE) within their respective promoters [4]. Reciprocally, PRR9/7/5 proteins also sequentially suppress the expression of *CCA1*/*LHY* from dawn to dusk [5,6]. In the evening, another important component is evening complex (EC), composed by EARLY FLOWERING 3 (ELF3), EARLY FLOWERING 4 (ELF4), and LUX ARRHYTHMO (LUX), which is able to act as transcriptional repression complex. ELF3 and ELF4 proteins localize in the nucleus and do not contain any of the identified functional domains so far [7,8]. LUX is a GARP transcription factor with a single MYB domain [9]. EC confers the nighttime repression to the clock by repressing the expression of *TOC1*, *GIGANTEA* (*GI*), and *PRR9* [9,10,11]. Loss of function any of the individual EC components will lead to circadian arrhythmia, indicating that EC plays a crucial role in maintaining the proper circadian clock activity [1,9,10]. F-box protein ZEITLUPE (ZTL), has been identified as a blue light receptor, containing light, oxygen, and voltage (LOV) domain at its N-terminus and tandem KELCH domain at its C-terminus, respectively [12]. ZTL plays an essential role in mediating the degradation of TOC1 and PRR5 at post-translation level [13,14]. Intriguingly, GI serves as a chaperone and interacts directly with both HEAT SHOCK PROTEIN (HSP90) and ZTL to form a ternary complex, thus, specifically facilitates the maturation of ZTL [15]. Additionally, GI recruits the deubiquitylases, UBP12 and UBP13, to regulate the accumulation of ZTL photoreceptor complex [16] (Figure 1). Circadian output pathways represent a plethora of downstream events regulated by, including the temporal regulation on plant growth and development, timing biotic and abiotic stresses, and modulation on multiple phytohormone signaling pathways. For instance, EC directly binds and represses the expression of PHYTOCHROME INTERACTING FACTOR 4 (*PIF4*) and *PIF5* to gate the hypocotyls growth in late night [10]. In the other hand, independent of EC, ELF3 alone can also regulate hypocotyl elongation by physically interacting with PIF4 protein to inhibit its transcriptional activity [17]. Importantly, circadian clock regulates biological processes mediated by hormones through affecting hormone biosynthesis, signaling, and response pathways, such as the defense hormones, salicylic acid (SA) and jasmonic acid (JA) [18,19].

JA has been well recognized as a plant defense related hormone, which mainly regulates plant response to biotic stresses, including herbivore and pathogen attack. JA also plays crucial roles in mediating various biological events, such as photomorphogenesis, root growth, leaf senescence, wounding response, regeneration, abiotic stress responses, herbivory, and pathogen infections [20,21,22,23,24,25]. The biosynthesis of JA has been well characterized [26,27,28]. In brief, bioactive JA, (+)-7-*iso*-JA-Ile (JA-Ile), is generated from a trienoic fatty acid through the octadecanoid pathways (Figure 2). JA metabolism pathways convert JA into active and inactive compounds (Figure 2). The JA perception shares canonical ubiquitin-proteasome system with other hormones, such as gibberellin (GA) and auxin. CORONATINE INSENSITIVE 1 (COI1), a F-box protein, acts as JA receptor, which can bind bioactive JA and trigger the formation of receptor complex COI1-JA-Ile-JAZ, to promote the ubiquitination and degradation of JASMONATE ZIM DOMAIN (JAZ) proteins [29,30,31,32]. JAZ family consists of 13 members in *Arabidopsis*, and most of them possess two conserved domains, namely Zn-finger protein expressed in inflorescence meristem (ZIM) and Jas domains [33]. ZIM domain is responsible to mediate the interaction with NOVEL INTERACTOR OF JAZ (NINJA) or dimerization of JAZ proteins themselves, while Jas domain facilitates its interaction with COI1 and other transcription factors [25,34,35]. To date, lots of JAZ targets have been identified, including MYC, MYB, NAC, ERF, and WRKY family transcriptional factors, which mediate various downstream JA responses. MYC2, a basic-helix-loop-helix (bHLH) transcription factor, has been considered as a master downstream regulator of JA signaling pathway. A recent study showed that MEDIATOR 25 (MED25), a subunit of the mediator coactivator complex, could bridge COI1 to RNA Polymerase II and chromatin, thus, to trigger JA signaling. MED25 physically interacts with COI1 and histone acetyltransferase 1 (HAC1), and cooperatively mediates the histone (H) 3 lysine (K) 9 acetylation (H3K9ac) modification within the promoters of MYC2 target genes [36] (Figure 2).

MYC2, together with MYC3 and MYC4, belongs to the basic-helix-loop-helix IIIe transcription factor family. They are direct targets of JAZ repressors to play critical roles in mediating various aspects of the JA response in *Arabidopsis* [37]. They redundantly regulate the activation of JA-induced leaf senescence, by binding *SENESCENCE-ASSOCIATED GENE 29* (*SAG29*)promoter thus to activate its expression [38]. Interestingly, members of bHLH subgroup IIId transcriptional factors including bHLH03, bHLH13, bHLH14, and bHLH17, can inhibit the functions of MYC2, MYC3, and MYC4 in regulating JA-induced leaf senescence [38]. In addition, MYC2, MYC3, and MYC4 can also directly bind and promote the transcription of *PHEOPHORBIDE A OXYGENASE* (*PAO*), *NON-YELLOWING 1* (*NYE1*), and *NON-YELLOW COLORING 1* (*NYC1*), which are associated with chlorophyll degradation during leaf senescence [39]. Altogether, these evidences demonstrated that the JA signaling pathway was involved in JA-mediated senescence. Furthermore, it is also reported that exogenous JA treatment repressed flowering time by inhibiting *FT* expression, partially through MYC2/3/4 [40]. Recently, it has been implicated that both JA homeostasis and signaling pathway are regulated by circadian clock, while JA is capable of regulating circadian speed in a feedback manner. Here we summarize the emerging crosstalks between circadian clock and JA pathway, and list out the perspective for future investigation on their crosstalk network, which might shed light on the tradeoff among plant growth, development, and defense to optimize plant growth and reproductive behavior.

## 2. Biosynthesis and Metabolism of JA are Regulated by Circadian Clock

Transcriptomic studies have shown many of the defense-associated genes are regulated by circadian clock [41,42,43,44]. Consistently, the accumulation of JA content displayed a well rhythmic oscillation pattern with the peak at middle of the subjective day time and the trough level at around the middle night, indicating that JA biosynthesis and homoeostasis might be regulated by circadian clock [45]. As expected, the transcriptional profile of *SULFOTRANSFERASE 2A* (*ST2A*) which encodes a sulfotransferase family protein to involve the metabolism of JA, is significantly up-regulated at the end of dark phase under short day, and controlled by circadian clock [46]. Moreover, transcriptomic profiling analysis displayed that *LIPOXYGENASE 3* (*LOX3*) and *LIPOXYGENASE*
*4 (LOX4*), encoding two 13-lipoxygenases which directly catalyze the biosynthesis of JA, were significantly up-regulated in *lux arrhythmo* (*lux*) mutant, which further indicated the biosynthesis of JA may be regulated by Evening Complex [47]. Collectively, both JA biosynthesis and metabolism genes are controlled by circadian clock with peak at specific time of the day, hence to cause the rhythmic JA accumulation pattern.

## 3. Circadian Clock Regulates JA-Mediated Plant Development Events

In animals, the circadian clock has been reported to be tightly associated with aging process, and the dysfunction or disruption of circadian clock will dramatically accelerate the aging process. Nevertheless, whether the circadian clock regulates the aging or senescence process in higher plants is still largely unknown. Intriguingly, circadian stress from the changed regime of light-dark duration results in the lesser transcript levels of *CCA1* and *LHY*. Strikingly, circadian stress, which changed the regime of light-dark duration, also causes dramatically JA-dependent cell death process in cytokinin deficient plants and clock related mutants [48]. Very recently, transcriptomic profiling analysis revealed that evening complex is involved in the regulation of JA signaling and response by timing *MYC2* transcription (Figure 3). Meanwhile, the JA content is decreased in EC mutant examined at midday, the peak time for JA accumulation, which may result from the feedback regulation of activated JA signaling pathway [47]. Time for coffee (TIC), a component of circadian clock, functions as a negative regulator of JA signaling pathway, and the JA responses to root length inhibition is defective in *tic* mutant. In this scenario, TIC protein can directly interact with MYC2 and inhibit its protein turnover specifically in the evening phase [49] (Figure 3). Thus, we concluded that, not only JA content, but also its signaling pathway are modulated by circadian clock, indicating the complex cross network between circadian clock and JA regulated cellular events.

## 4. Circadian Clock Gates JA Regulated Herbivory Resistance

Circadian clock confers the ability of plant to anticipate diel abiotic threats including herbivory and pathogen resistance, while JA is one of the major hormones during this process. The feeding behavior of cabbage loopers, *Trichoplusia*
*ni* (*T. ni*), is rhythmic under constant conditions [45]. Plants entrained in-phase with the insects display much more resistance to the insect-attacking, with less tissue damage [45]. These phenotypes indicated that the feeding behavior of insects could be under the control of circadian clock. Furthermore, the circadian arrhythmic plants, *lux* mutant and *CCA1-OX* plants, did not display drastic difference in plant tissue loss compared with wild type, when challenged with in- and out-of-phase entrained *T. ni*. This finding implies that circadian clock is required for plant defense against herbivory. The plant tissue loss of *allene oxide synthase* (*aos*)and *jasmonate resistant 1* (*jar1*) mutants, in which the biosynthesis of JA is defective, also show no obvious difference when treated with in- and out-phase entrained *T. ni.* Both the circadian and JA mutants fail to display enhanced *T. ni* resistance, even entrained in-phase to insect, implying that both circadian clock and JA pathway are required for the in-phase-dependent enhanced herbivory resistance [45]. When plants encounter herbivores, it will emit complex volatile compounds to against this biotic stress, of which green leaf volatiles (GLVs) are a kind of fatty acid-derived compounds emitted upon plant damage [50]. *HYDROPEROXIDE LYASE* (*HPL*), encoding a GLV biosynthetic enzyme, is regulated by circadian clock at transcriptional level in *Nicotiana attenuata*. Accordingly, the emission of GLV is also rhythmic, with a peak at midday while its trough level at night. Moreover, JA signaling increases the basal turnover of *NaHPL* transcripts. Taken together, the GLV emissions are co-regulated by damage, JA signaling and circadian clock [51]. A study showed that the internal floral rhythm is abrogated in *NaLHY* and *NaZTL* RNA interference transgenic lines in *Nicotiana attenuata* [52]. The *NaZTL* RNAi transgenic lines are more susceptible to generalist herbivore *Spodoptera littoralis* compared to wild type. Plants usually produce various secondary metabolites to defense herbivores [53]. Nicotine is one of the most efficient defense-related metabolites in *Nicotiana attenuata*. To investigate whether the accumulation of nicotine confers to the attenuated *Spodoptera littoralis* resistance in *NaZTL* RNAi line, the secondary metabolites levels were measured. The results displayed that the nicotine level was significantly decreased in *NaZTL* RNAi transgenic lines than in the control plants. Furthermore, exogenous supplementation of nicotine could rescue the attenuated resistance to *Spodoptera littoralis* in *NaZTL* RNAi transgenic lines [54]. These findings suggested that the nicotine levels mediate the resistance against to *Spodoptera littoralis* in *NaZTL* RNAi transgenic lines. Moreover, they also found that the transcriptional levels of nicotine biosynthesis genes were significantly decreased in *NaZTL* RNAi plants. Meanwhile, the biosynthesis of nicotine is also mediated by JA signaling. NaZTL interacts with JASMONATE ZIM domain (JAZ) protein in the COI1 dependent manner, thus mediating JA signaling to gate plant defense response. All these evidences showed that *NaZTL* RNAi plants were more susceptible to *Spodoptera littoralis*, partially due to the reduced JA-regulated accumulation of nicotine in *Nicotiana attenuata* [54]. Taken together, all these evidences implied the essential roles of circadian clock in gating JA regulated herbivory resistance.

## 5. Circadian Clock Gates Temporal Variation Susceptibility to Pathogen by Jasmonates

Besides of gating herbivory resistance, circadian clock is also playing a vital role in gating JA mediated-defense against to pathogen. CCA1, a single MYB-domain containing transcriptional repressor with peak expression at dawn, has been identified as a regulator of plant defense [41]. TIC has also been reported to involve the circadian clock gated pathogen defense [49]. To address whether there is a temporal variation susceptibility to pathogen in *Arabidopsis*, Col-0 plants were challenged with pathogen *Pseudomonas syringae* pv. *tomato* (*Pst*) DC3000 at different circadian times under constant light conditions to avoid the effect of light/dark photoperiod. It turned out that Col-0 is more susceptible to *Pst DC3000* at subjective midnight than at subjective morning [55]. However, the temporal variation susceptibility to pathogen was vanished in two circadian arrhythmic plants, namely *CCA1*-overexpressing line (*CCA1-ox*) and *elf3* mutant. Expression profiling analysis further revealed that a series of known defense related genes, including components of SA-related signaling pathway, namely *ISOCHORISMATE SYNTHASE 1* (*I**CS1*), *ENHANCED DISEASE SUSCEPTIBILITY 1* (*EDS1*), *ENHANCED DISEASE SUSCEPTIBILITY 5* (*EDS5*)*,* and CONSTITUTIVE EXPRESSION OF PR GENE 5 (*CPR5*) and the receptor of jasmonic acid (*COI1*), are transcriptionally regulated by circadian clock in *Arabidopsis* [55]. These lines of evidences suggest that the temporal variation susceptibility to pathogen is indeed under the control of circadian clock [55]. In addition, Korneli and colleagues show that circadian clock controls the discrepancy of pre-invasive and post-invasive defense responses against pathogens. They found that the distinct time of day responses to pathogen in another arrhythmic mutant *lux* (*lux arrhythmo*), such as oxidative burst and cell death, is different to those in wild type [56]. Very recently, the interplay between circadian clock and JA signaling was further elaborated. *LUX* was found to be induced when challenging with pathogen, and *LUX* is partially involved into the stomata-dependent defense response. Transcriptomic profiling and ChIP-seq analysis characterized *EDS1* and *JAZ5* as the novel targets of LUX in the regulation of JA signaling (Figure 3) [57]. Taken together, the above evidences clearly indicated the prominent role of circadian clock in gating plant defense, especially in a JA signaling dependent manner.

## 6. Circadian Clock is Associated with the Crosstalk between JA and SA Signaling

JA, together with SA, has been characterized as defense associated hormones. Intriguingly, the JA and SA signaling display antagonistic roles to each other in many defense processes, and their abundance are usually reverse to each other [1,58]. So far, several studies have indicated that the circadian clock delicately primes the JA and SA signaling. The peak of SA is at the nighttime while JA is at the midday, which is associated with the defense against morning biotrophic and dusk herbivore attacks respectively [58]. *PHT4;1* has been previously shown to affect SA-mediated defense. Wang and colleagues found that the expression of phosphate transporter gene *PHT4;1* was under the control of CCA1, and CCA1 could directly bind the promoter of *PHT4;1* to shape its diel transcription pattern, thus mediating SA dependent defense resistance [59]. ICS1 is a central enzyme for biosynthesis of SA, its expression can be activated upon various pathogens challenges. Yeast one hybrid assay found that CCA1 HIKING EXPEDITION (CHE) protein could directly bind *ICS1* promoter. Further study found that the circadian expression patterns of *CHE* and *ICS1* were similar. Moreover, the expression of *ICS1* was reduced in *che-2* mutant. Consistently, the SA accumulation was also decreased in *che-2*. These findings suggested that CHE served as an activator of *ICS1* and SA contents (Figure 3) [60]. It seems like that the timing SA and JA oscillations at different time window by circadian clock confers plant temporal variation susceptibility to specific invaders. Meanwhile, this separation of JA and SA may be able to avoid their potential antagonism [58]. Plant immunity usually causes the alteration of the cellular redox state including the total level of glutathione, the ratio of reduced (GSH) and oxidized (GSSG) forms of glutathione. SA increases the ratio of GSH/GSSG, by contrast, JA decreases the ratio of GSH/GSSG and the accumulation of glutathione. Unsurprisingly, glutathione is also involved in the regulatory crosstalk between JA and SA [58,61,62]. *Arabidopsis* NON-EXPRESSOR OF PATHOGENESIS-RELATED GENE 1 (NPR1) functions as a redox state sensor and regulates transcription of core circadian clock genes such as *PRR7* and *TOC1* [58]. SA triggers the enhanced redox status, increased glutathione accumulation and GSH/GSSG ratio, which resulted in the reinforcement of circadian clock [58] (Figure 3). Altogether, this balanced regulatory loop may maximize plants adaptability to external environment.

## 7. JA Signaling Reciprocally Affects Clock Activity

It has been shown that many of plant hormones biosynthesis and signaling are under the control of circadian clock to exert their function in specific time of day. Nonetheless, whether and how phytohormones regulate clock activity such as circadian phase and period, amplitude had been rarely explored. In higher plants, there are nine kinds of well-known hormones including auxin, gibberellins, cytokinin, abscisic acid (ABA), ethylene, SA, JA, brassinosteroide (BR), and strigolactones (SL) [63,64,65,66]. Years ago, the phytohormone effects on circadian clock have been examined using pharmaceutical treatment and hormone related mutants. They found that auxin could regulate circadian amplitude and clock precision, cytokinins could delay circadian phase, while brassinosteroid and abscisic acid could modulate circadian period. In contrast, gibberellins and ethylene had no effects on circadian clock [67]. SA has been experimentally shown to modulate the amplitude of circadian clock accompanied by the redox rhythm [68]. As for JA, recent study suggested that JA signaling also could reciprocally affect clock activity, and the expression of a few core components of circadian clock, such as *CCA1*, *LUX,* and *GRP7*, were reduced upon JA treatment. Further, JA treatment also resulted in a dampened amplitude of *CCA1:LUC* [57]. Moreover, the circadian period could be significantly lengthened in Col-0 by treated with JA-isoleucine (JA-Ile), a kind of bioactive JA derivative [57]. However, the underlying mechanism of JA treatment caused the lengthened circadian period still remains unclear (Figure 3). In the future, it would be required to systematically investigate the circadian phenotypes of JA signaling related mutants to pinpoint the exact molecular links from JA signaling to circadian clock, which might be occurred at transcriptional and post-transcriptional levels, or both.

## 8. Perspectives

To battle against various pathogens and pests, sessile plants had evolved many conserved and sophisticated defense mechanisms. Circadian clock is an internal time-keeping mechanism, which confers the anticipation of plants to the surrounding environmental cues such as daily changing light and temperature information. The emerging studies indicate that circadian clock play crucial roles in mediating leaf senescence. Evening complex, constituted by ELF4-ELF3-LUX, can directly bind and repress the expression of *MYC2*, which is a master regulator for JA signaling pathway, thus gating the circadian output regulation on JA-induced leaf senescence [47]. Another recent study confirmed that Evening Complex mutants showed hypersensitivity to dark-induced leaf senescence [69]. By contrast, the dark-induced leaf senescence is significantly delayed in *prr9* mutant, as PRR9 can directly bind the promoter of *ORESARA1* (*ORE1*) and activate its transcription. Genetically, *ORE1* overexpression can rescue the delayed leaf senescence of *prr9*. Collectively, circadian component PRR9 serves as a novel regulator of leaf senescence via positively activation of *ORE1* [69]. Reciprocally, circadian period is also feedback regulated by the leaf aging process. The circadian periods are about 1 h shorter in older leaves than in younger leaves, implying that aging process is associated with the regulation of circadian period [69]. Further study showed that TOC1 plays a central role in linking age to circadian clock period regulation [69]. Intriguingly, as aging, the JA response decays in *Arabidopsis*, which is regulated by the interaction between SQUAMOSA PROMOTER BINDING PROTEIN-LIKE 9 (SPL9) and JAZ protein [70]. Whether SPL directly mediate the circadian period needs to be further investigated in the future. Recent study shows that JA signaling reciprocally affects clock activity, however, its elaborated mechanism still remains elusive. The mechanism of JA signaling coupled to circadian clock is warranted to be further investigated by using JA signaling and circadian clock related mutants. Moreover, whether the regulation of JA on circadian activity is direct or indirect is still unknown. The network and the links between JA and circadian clock need to be further disclosed.

Besides of regulation on leaf senescence and defense resistance, JA also plays essential roles in regulating photomorphogenesis, root growth, photoperiodic dependent flowering time control, abiotic stress response and sterility as well [26]. Whether circadian clock gates these outputs and orchestrates the interaction between various JA responses, especially the tradeoff between growth and defense, awaits to be future investigated.

Furthermore, it has been known the biosynthesis of JA is circadian clock regulated and with a peak at midday [45], however, its underlying molecular mechanism is still unknown yet. The biosynthesis of JA is extremely complex, and a series of enzymes are involved in. Grundy and colleagues proposed that TOC1, PRR5, and PRR7 may be the novel regulators of both JA signaling and biosynthesis, due to the binding of these proteins to the promoters of JA responses genes by ChIP-seq analysis, including *LOX2*, *LOX3*, *LOX4*, *JAZ1*, *JAZ9*, *OCP3*, *PFT1*, *WRKY40*, *MYB108,* and *ANAC019* [71]. It would be interesting to know whether these genes are regulated by circadian clock. Further investigation with big-data driven approaches including transcriptomics, together with classical biochemical assays including protein-protein interaction, protein-DNA interaction, and genetic tools might be able to reveal this mask.

JA is usually produced when attacked and regulates inducible defenses. However, multiple lines of evidence showed that JA was regulated by circadian clock, which strongly suggested a more constitutive role of circadian clock in gating plant abiotic stress. Circadian clock confers the ability of plant to anticipate diel abiotic threats including herbivory and pathogen resistance at specific time of day, thus to enhance fitness of plants. However, the reciprocal regulation mechanisms among circadian clock, JA signaling, and abiotic stress are still not clear, which need to be further disclosed.

## Figures and Tables

**Figure 1 ijms-20-05254-f001:**
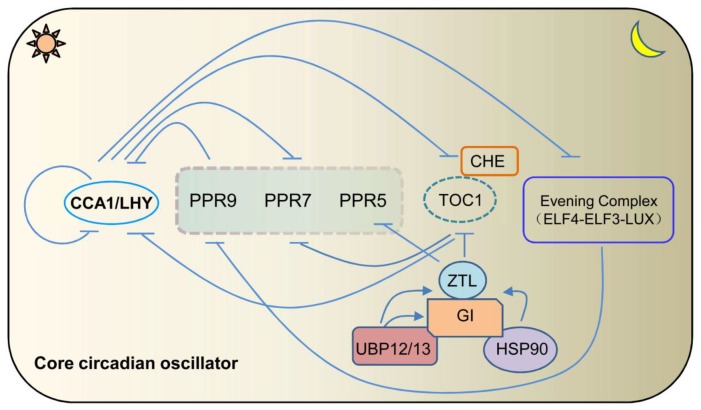
A simplified model for the circadian clock in *Arabidopsis thaliana*. Morning expressed CIRCADIAN CLOCK-ASSOCIATED1 (CCA1) and LATE ELONGATED HYPOCOTYL (LHY) repress the expression of all of the *PRR* family members and evening complex (EC). All the PSEUDO-RESPONSE REGULATOR (PRRs) reciprocally repress the expression of *CCA1* and *LHY*. EC is composed by EARLY FLOWERING 4 (ELF4), EARLY FLOWERING 3 (ELF3), and LUX ARRHYTHMO (LUX), and acts as a negative regulator of *PRR9*. GIGANTEA (GI) acts as a co-chaperone, recruiting HSP90 for the maturation of the ZEITLUPE (ZTL) protein. Additionally, GI recruits the deubiquitylases, UBP12 and UBP13, to regulate the accumulation of ZTL photoreceptor complex.

**Figure 2 ijms-20-05254-f002:**
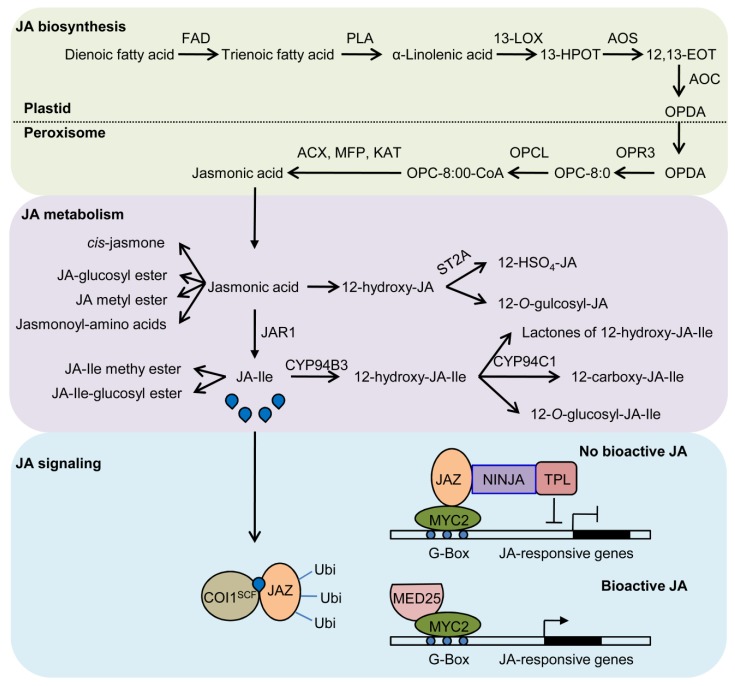
A model for jasmonic acid (JA) biosynthesis, metabolism and signaling pathways. JA-Ile is generated from trienoic fatty acid through the octadecanoid pathways. JA metabolism pathways can convert JA into active or inactive compounds. Coronatine insensitive1 (COI1), a F-box protein, acts as JA receptor, which can bind bioactive JA to trigger the formation of receptor complex COI1-JA-Ile-JAZ, hence to promote the ubiquitination and degradation of jasmonate zim domain (JAZ) proteins, then resulting in the release of downstream transcription factors, such as MYC2, and activation of JA responsive genes. ST2a, 12-OH-JA sulfotransferase; FAD, fatty acid desaturase; PLA, phospholipase A1; 13-LOX, 13-lipoxygenase; 13-HPOT, 13-hydroperoxyoctadecatrienoic acid; AOS, allene oxide synthase; 12,13-EOT, 12,13(S)-epoxyoctadecatrienoic acid; AOC, allene oxide cyclase; OPDA, (9S,13S)-12-oxo-phytodienoic acid; OPR, OPDA reductase. OPC-8:0, 3-oxo-2(cis-2’-pentenyl)-cyclopentane-1-octanoic acid; OPCL, OPC-8:0 CoA ligase; ACX, acyl-CoA oxidase; KAT, 3-ketoacyl-CoA thiolase; MFP, multifunctional protein; JAR1, jasmonate resistant 1; JA-Ile, jasmonoyl-l-isoleucine.

**Figure 3 ijms-20-05254-f003:**
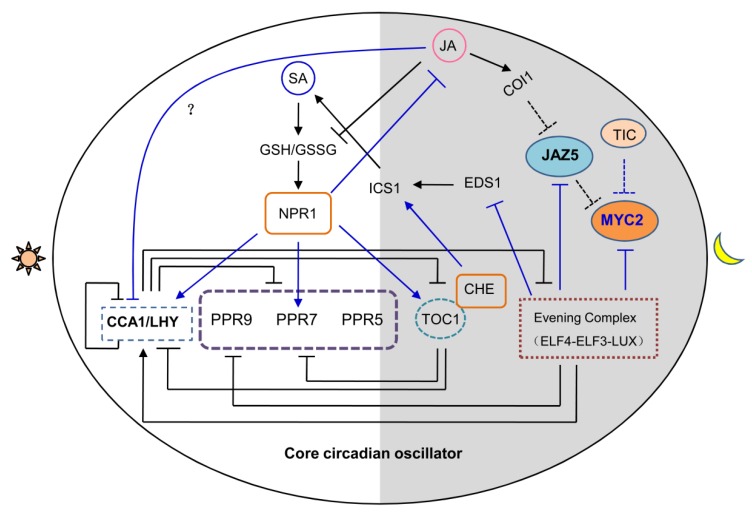
A proposed model for crosstalks between circadian clock and JA pathway in *Arabidopsis*. Evening complex (EC) transcriptionally represses the expression of *MYC2*, an essential master of JA signaling pathway to mediate leaf senescence. *EDS1* and *JAZ5* are the direct targets of LUX in regulating salicylic acid (SA) and JA signaling. Time for coffee (TIC) directly interacts with MYC2 and inhibits MYC2 protein turnover by timing the transcriptional level of CORONATINE INSENSITIVE 1 (COI1) in an evening-phase-specific manner. CCA1 hiking expedition (CHE) serves as an activation of ISOCHORISMATE SYNTHASE 1 (*ICS1*), an enzyme essential for SA biosynthesis. SA modulates the amplitude of circadian clock accompany by the redox rhythm and NON-EXPRESSOR OF PATHOGENESIS-RELATED GENE 1 (NPR1). NPR1 regulates the expression of circadian genes, *CCA1*, *LHY*, *PRR7,* and *TIMING OF CAB EXPRESSION 1* (*TOC1*). Reciprocally, JA signaling affects clock activity through unknown mechanisms. Novel molecular links between circadian clock and JA signaling pathway await to be further investigated.

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
