# Peer review of "Novel Crosstalks between Circadian Clock and Jasmonic Acid Pathway Finely Coordinate the Tradeoff among Plant Growth, Senescence and Defense"

_ijms, 2019, doi:10.3390/ijms20215254_

Round 1

Reviewer 1 Report

The manuscript by Zhang et al. attempts to describe the effects of the circadian clock on jasmonic acid (JA)- induced responses in plants. The manuscript is well organized and examples have been properly selected. Unfortunately, there are also many issues with the manuscript leading to my recommendation.

First, the authors should revisit their use of articles (the, a, an) in the text. They are rarely used at all. In general, the manuscript requires some massive english editing.

Past tense should be used throughout the text.

Secondly, there are several statements that need clarification or alterations. In line 65, the authors list biological events, but omit herbivory and pathogen infections, which are also examples for biological events.

In lines 67 and 68, the biosynthetic pathway for JA is briefly described. However, JA is made from a trienoic fatty acid (not dienoic), and is not modulated through a series of oxylipin biosynthetic pathways, but rather through the octadecanoid signalign pathways. I am aware that there might be an alternative pathway for some, but then it should be described as such.

In line 69, GA should be written out.

Line 102, what does "by binding and activation of SAG29 expression" mean? This sentence should be reworded for clarification.

Lines 107, 108; It seems obvious that the JA signaling pathway is involved in JA-mediated senescence. Sentence needs rewording.

Lines 113, 114, "which might configure..." should be reworded.

Line 122;  second part of sentence ("could be critically induced....") is unclear and needs rewording.

Line 125; This is the first time lux is mentioned and should therefore be explained.

Line 131; "will dramatically facilitate..." needs clarification.

Lines 137, 138; Description of EC is redundant since is was described before.

Lines 148, 149; "could" does not "indicate" because it is speculation. Sentence needs rewording.

Lines 162-169; This whole segment is not clear to the reader who is not familiar with the topic and needs to be better organized.

Lines 177, 178; Herbivores are per definition enemies of plants; redundant

Line 198; "All these evidences collectively indicate..". No, they show that it is.

Lines 212, 213; "at different circadian time (should be "times"), under constant light conditions.." Either no comma or, if both were used, clarification.

Line 226; What is stomata-dependent defense?

Lines 232, 232; SA is not a major herbivore defense signal

Lines 234, 235; 'and their abundance are reversely accumulated with each other" should be reworded.

Line 237; "responsible" is a strong word here in this context and suggests that those are the major events regulating defenses, which is not true. 

Line 266; JA should be listed just as JA, not jasmonic acid, since it was introduced previously.

The whole "Perspectives" section needs re-writing. Right now it is mainly a summary with very little outlook or perspective. What I would also like to see is some discussion of the relevance of JA being at least partially under the control of a clock. This aspect is totally missing. However, JA is being produced when attacked and regulates inducible defenses. Being regulated by a clock strongly suggests a more constitutive role. And do all herbivores have a circadian patterns or do pathogens only attack in the morning? These aspects should also be discussed.

Author Response

The manuscript by Zhang et al. attempts to describe the effects of the circadian clock on jasmonic acid (JA)- induced responses in plants. The manuscript is well organized and examples have been properly selected. Unfortunately, there are also many issues with the manuscript leading to my recommendation.

First, the authors should revisit their use of articles (the, a, an) in the text. They are rarely used at all. In general, the manuscript requires some massive english editing. Past tense should be used throughout the text.

Reply: Thank you for your suggestion. We have checked all the articles, punctuations throughout the main text carefully, and carefully edited the revised manuscript.

Secondly, there are several statements that need clarification or alterations. In line 65, the authors list biological events, but omit herbivory and pathogen infections, which are also examples for biological events.

Reply: Thanks for your helpful suggestion. We have added the following words in the revision version: “JA also plays crucial roles in mediating various biological events, such as photomorphogenesis, root growth, leaf senescence, wounding response, regeneration, abiotic stress responses, herbivory and pathogen infections etc.”.

In lines 67 and 68, the biosynthetic pathway for JA is briefly described. However, JA is made from a trienoic fatty acid (not dienoic), and is not modulated through a series of oxylipin biosynthetic pathways, but rather through the octadecanoid signaling pathways. I am aware that there might be an alternative pathway for some, but then it should be described as such.

Reply: Sorry for the misleading. We have changed this in the revision as follows: “is generated from a trienoic fatty acid through the octadecanoid pathways (Figure 2).”

 In line 69, GA should be written out.

Reply: As suggested, gibberellin (GA) have been spelled out.

Line 102, what does "by binding and activation of SAG29 expression" mean? This sentence should be reworded for clarification.

Reply: Sorry for the misleading. This statement have changed in the revision as “by binding the promoter of SAG29 thus to activate its expression”.

Lines 107, 108; It seems obvious that the JA signaling pathway is involved in JA-mediated senescence. Sentence needs rewording.

Reply: Sorry for the misleading. This sentence changed as “Altogether, these evidences demonstrated that the JA signaling pathway is involved in JA-mediated senescence”.

Lines 113, 114, "which might configure..." should be reworded.

Reply: As you suggested, "which might configure..." reworded as “might shed light on”.

Line 122;  second part of sentence ("could be critically induced....") is unclear and needs rewording.

Reply: Sorry for the misleading. "could be critically induced...." was reworded as “is significantly up-regulated”.

Line 125; This is the first time lux is mentioned and should therefore be explained.

Reply: As you suggested, “lux” have been changed as “lux arrhythmo (lux)” in the revision.

Line 131; "will dramatically facilitate..." needs clarification.

Reply: We have changed it into “will dramatically accelerate”.

Lines 137, 138; Description of EC is redundant since is was described before.

Reply: As you suggested, we deleted this in the revision.

Lines 148, 149; "could" does not "indicate" because it is speculation. Sentence needs rewording.

Reply: Sorry for the misleading. We have reworded as “Thus, we conclude that, not only JA content, but also its signaling are modulated by circadian clock, indicating the complex cross network between circadian clock and JA regulated cellular events.”

Lines 162-169; This whole segment is not clear to the reader who is not familiar with the topic and needs to be better organized.

Reply: As you suggested, we simplified this part as follows: “Plants entrained in-phase with the insects display much more resistance to the insect-attacking, with less tissue damage.”

Lines 177, 178; Herbivores are per definition enemies of plants; redundant

Reply: As you suggested, we changed this as “When plants encounter herbivores”.

Line 198; "All these evidences collectively indicate..". No, they show that it is.

Reply: As you suggested, we changed "All these evidences collectively indicate.." as “All these evidences showed that”. Thanks.

Lines 212, 213; "at different circadian time (should be "times"), under constant light conditions.." Either no comma or, if both were used, clarification.

Reply: As you suggested, we rephrased it as “at different circadian times under constant light conditions”.

Line 226; What is stomata-dependent defense?

Reply: Sorry for the misleading, in the original reference, they report that: “For epiphytic bacterial pathogens, such as spray-infected P. syringae, plants show higher susceptibility in the morning than at night. Because epiphytic bacteria need to pass through stomata to gain access to the interior of plant tissue and the infiltrated bacteria bypass this physical barrier, the differential resistance of plants to pathogens with different infection modes suggests that stomata-independent defense is strong during the day while stomata-dependent defense is dominant at night.”

Zhang, C.; Gao, M.; Seitz, N. C.; Angel, W.; Hallworth, A.; Wiratan, L.; Darwish, O.; Alkharouf, N.; Dawit, T.; Lin, D.; Egoshi, R.; Wang, X.; McClung, C. R.; Lu, H., LUX ARRHYTHMO mediates crosstalk between the circadian clock and defense in Arabidopsis. Nat Commun 2019, 10, (1), 2543.

Lines 232, 232; SA is not a major herbivore defense signal

Reply: As you suggested, we deleted “predominantly functioned in herbivore resistance and response to pathogen attack” in the revision.

Lines 234, 235; 'and their abundance are reversely accumulated with each other" should be reworded.

Reply: As you suggested, this have been reworded as: “and their abundance are usually reverse to each other”

Line 237; "" is a strong word here in this context and suggests that those are the major events regulating defenses, which is not true. 

Reply: As suggested, we have changed “responsible” to “is associated with the”

Line 266; JA should be listed just as JA, not jasmonic acid, since it was introduced previously.

Reply: Thank you for your suggestion. We used JA in the revision.

The whole "Perspectives" section needs re-writing. Right now it is mainly a summary with very little outlook or perspective. What I would also like to see is some discussion of the relevance of JA being at least partially under the control of a clock. This aspect is totally missing. However, JA is being produced when attacked and regulates inducible defenses. Being regulated by a clock strongly suggests a more constitutive role. And do all herbivores have a circadian patterns or do pathogens only attack in the morning? These aspects should also be discussed.

Reply: Thank you for your suggestion. We discussed this part as follows: “JA is usually produced when plant is attacked and hence to regulate the inducible defenses. However, multiple lines of evidence show that JA is regulated by circadian clock, which strongly suggests a more constitutive role of circadian clock in gating plant abiotic stress. Circadian clock, an internal timekeeping mechanism, confers the ability of plant to anticipate diel abiotic threats including herbivory and pathogen resistance at specific time of the day, thus to enhance plant fitness. However, the reciprocal regulation mechanisms among circadian clock, JA signaling and abiotic stress are still not clear, which need to be further disclosed.”

Reviewer 2 Report

This review examines the growing body of literature of crosstalk between the circadian clock and jasmonate defenses, but with a unique perspective on JA-mediated regulation of growth and development. The manuscript provides good detail while synthesizing new perspectives, though at times these may be too speculative. I have only a few minor suggestions to improve the manuscript.

Suggestions:

- Due to the complex nature of the circadian clock, a simplified schematic might be helpful for the introduction.

- ST2a is referred to as a component of JA metabolism and, considering that the turnover of bioactive hormone is an active area of research and that the rest of JA biosynthesis is very detailed, it may be worthwhile to include JA catabolism as part of Fig 1

- Revise section 4 to improve clarity of circadian gating of herbivore resistance. A solution could be a better explanation of in-phase and out-of-phase entrainment or simplification of how the clock gates herbivore resistance.

- In perspectives, the idea that JA modulation of the clock explains age-dependent changes in the clock is too speculative. In the absence of environmental or developmental signals, there should be no JA signal to influence the clock. Simpler mechanisms may explain an age-dependent change to the clock (e.g., direct SPL interaction with clock).

- The rationale for describing JA signaling in tomato and Ficus on lines 328-336 of the perspectives appears to be to demonstrate that JA signaling is conserved across species. This paragraph is awkward and largely unnecessary. However, if the authors want to illustrate the conservation of JA signaling, recent studies describing OPDA, COI1, JAZ, and MYC in Marchantia might to better for this purpose.

- While understandable, the manuscript would benefit by revision for English.

Minor comments:

- line 35: change “finding” to “founding”

- line 68: suggest referring to Fig. 1 when first describing JA biosynthesis – common theme; references are often not cited at first mention – when not cited immediately, it becomes unclear to the reader what literature is used as evidence

- line2 133-136: Please define circadian stress.

- JA-Ile in Fig.1 should be between COI1 and JAZ, for which JA-Ile acts as a molecular glue; current schematic suggests that JA-Ile binds only COI1

- lines 209-210: revise wording suggesting that TIC influences MYC2 protein turnover by regulating expression of COI1, since these processes (although correlated) are not mechanistically linked

- Can the different mechanisms of clock-gating of MY2 activity be described in the same section instead of two separate sections (3 & 5)? Also please cite the manuscript that MYC2 is controlled by the evening complex when describing this.

- lines 165-167: Sentences are redundant: less tissue damage is the same as more tissue remaining; remove sentence on line 166-167.

- line 213: change “photocycle” to “photoperiod” (these terms are not synonymous)

Author Response

This review examines the growing body of literature of crosstalk between the circadian clock and jasmonate defenses, but with a unique perspective on JA-mediated regulation of growth and development. The manuscript provides good detail while synthesizing new perspectives, though at times these may be too speculative. I have only a few minor suggestions to improve the manuscript.

Suggestions:

- Due to the complex nature of the circadian clock, a simplified schematic might be helpful for the introduction.

Reply: We appreciate your suggestion, and have added a simplified model of circadian clock as the new Figure 1 in the revised manuscript.

- ST2a is referred to as a component of JA metabolism and, considering that the turnover of bioactive hormone is an active area of research and that the rest of JA biosynthesis is very detailed, it may be worthwhile to include JA catabolism as part of Fig 1

Reply: We appreciate your suggestion and have added JA metabolism pathway into Figure 2 in the revision.

- Revise section 4 to improve clarity of circadian gating of herbivore resistance. A solution could be a better explanation of in-phase and out-of-phase entrainment or simplification of how the clock gates herbivore resistance.

Reply: We appreciate your suggestion and have simplified this part in the revision.

- In perspectives, the idea that JA modulation of the clock explains age-dependent changes in the clock is too speculative. In the absence of environmental or developmental signals, there should be no JA signal to influence the clock. Simpler mechanisms may explain an age-dependent change to the clock (e.g., direct SPL interaction with clock).

Reply: We appreciate your suggestion and have added comment as “Whether SPL directly mediate the circadian period needs to be further investigated.”

- The rationale for describing JA signaling in tomato and Ficus on lines 328-336 of the perspectives appears to be to demonstrate that JA signaling is conserved across species. This paragraph is awkward and largely unnecessary. However, if the authors want to illustrate the conservation of JA signaling, recent studies describing OPDA, COI1, JAZ, and MYC in Marchantia might to better for this purpose.

Reply: We appreciate your suggestion and have deleted this section.

- While understandable, the manuscript would benefit by revision for English.

Reply: We have polished the English throughout the main text.

Minor comments:

- line 35: change “finding” to “founding”

Reply: We have changed it as suggested. Thanks!

- line 68: suggest referring to Fig. 1 when first describing JA biosynthesis – common theme; references are often not cited at first mention – when not cited immediately, it becomes unclear to the reader what literature is used as evidence

Reply: Thank you for your suggestion. We have referred to it immediately as you suggested.

- line2 133-136: Please define circadian stress.

Reply: Thank you for your suggestion. We defined circadian stress in the revision as “circadian stress, which changed the regime of light-dark duration,”.

- JA-Ile in Fig.1 should be between COI1 and JAZ, for which JA-Ile acts as a molecular glue; current schematic suggests that JA-Ile binds only COI1

Reply: Thank you for your suggestion. We changed the site of JA-Ile, which is between COI1 and JAZ.

- lines 209-210: revise wording suggesting that TIC influences MYC2 protein turnover by regulating expression of COI1, since these processes (although correlated) are not mechanistically linked

Reply: Thank you for your suggestion. We changed “inhibit its protein turnover by timing the transcriptional level of COI1 in an evening-phase-specific manner” to “inhibit its protein turnover in an evening-phase-specific manner”.

- Can the different mechanisms of clock-gating of MY2 activity be described in the same section instead of two separate sections (3 & 5)? Also please cite the manuscript that MYC2 is controlled by the evening complex when describing this.

Reply: Thank you for your suggestion. We moved this part to the section 3.

- lines 165-167: Sentences are redundant: less tissue damage is the same as more tissue remaining; remove sentence on line 166-167.

Reply: Thank you for your suggestion. We simplified this part as follows: “Plants entrained in-phase with the insects display much more resistance to the insect-attacking, with less tissue damage.”

- line 213: change “photocycle” to “photoperiod” (these terms are not synonymous)

Reply: Thank you for your suggestion. We have changed “photocycle” to “photoperiod” in the revision.

Reviewer 3 Report

The submitted manuscript entitled ‘Novel crosstalk between circadian clock and jasmonic acid pathway finely coordinates the tradeoff among plant growth, senescence and defense’ is a well written review paper, which summarize selected the most important studies on role of Jasmonic Acid (JA) and circadian clock published in the last several years. Due to the high importance of JA in adaptation to drought, maybe this topic can be more extended. It is very current problem in the light of climate change and this extension could increase the number of readers of this article. Generally, manuscript is very well written, understanding and interesting. Figures are informative and nicely prepared. The conclusions are logical and suggest the need for further research. Between lines 296-312 literature citations are missing. Before publishing in IJMS should be also checked for typos and punctuation errors.

Author Response

The submitted manuscript entitled ‘Novel crosstalk between circadian clock and jasmonic acid pathway finely coordinates the tradeoff among plant growth, senescence and defense’ is a well written review paper, which summarize selected the most important studies on role of Jasmonic Acid (JA) and circadian clock published in the last several years. Due to the high importance of JA in adaptation to drought, maybe this topic can be more extended. It is very current problem in the light of climate change and this extension could increase the number of readers of this article. Generally, manuscript is very well written, understanding and interesting. Figures are informative and nicely prepared. The conclusions are logical and suggest the need for further research. Between lines 296-312 literature citations are missing. Before publishing in IJMS should be also checked for typos and punctuation errors.

Reply: We appreciate your positive and encouraging comments. We have added the missed literature citations, and carefully checked the language throughout the manuscript, including the usage of punctuations. Thank you very much.

Round 2

Reviewer 1 Report

The manuscript has improved significantly, but still needs some minor language editing. While some issues were resolved there is still a need for this to make the text ready for publication.

Otherwise, I find the manuscript well organized with a detailed description of events that link the circadian clock with JA-related defenses. The Outlook section is much better now and does no longer just summarize the main text. 

Author Response

Comments and Suggestions for Authors

The manuscript has improved significantly, but still needs some minor language editing. While some issues were resolved there is still a need for this to make the text ready for publication.

Otherwise, I find the manuscript well organized with a detailed description of events that link the circadian clock with JA-related defenses. The Outlook section is much better now and does no longer just summarize the main text.

Reply: We appreciate your positive and encouraging comments. In the updated version, we have carefully checked the language throughout the manuscript. A few changes have been highlighted with blue letters. Thank you very much.